# Prediction of Cyanotoxin Episodes in Freshwater: A Case Study on Microcystin and Saxitoxin in the Lobo Reservoir, São Paulo State, Brazil

Munique de Almeida Bispo Moraes [1], Raphaella de Abreu Magalhães Rodrigues [1], Raju Podduturi [2], Niels Ole Gerslev Jørgensen [2,*] and Maria do Carmo Calijuri [1]

[1] Department of Hydraulics and Sanitation, São Carlos School of Engineering, University of São Paulo, 400 Trabalhador São Carlense Avenue, São Carlos 13566-590, Brazil; muniquebio@gmail.com (M.d.A.B.M.); raphaella.magalhaes@gmail.com (R.d.A.M.R.); calijuri@sc.usp.br (M.d.C.C.)

[2] Department of Plant and Environmental Sciences, University of Copenhagen, Thorvaldsensvej 40, 1871 Frederiksberg C, Denmark; raju@food.ku.dk

[*] Correspondence: nogj@plen.ku.dk

**Abstract:** Freshwater reservoirs constitute an important source of drinking water, but eutrophication and higher temperatures increase the risk of more frequent blooms of cyanobacteria, including species that produce toxins. To improve the prediction of cyanotoxin episodes, we studied the annual occurrence of potential microcystin (MC) and saxitoxin (STX)-producing cyanobacteria in the Lobo reservoir, São Paulo State, Brazil. Relationships among environmental variables, cyanobacterial biomass, numbers of the *mcyE* and *sxtA* genotypes (genes encoding production of MC and STX, respectively), and concentrations of MC and STX were determined to address variables applicable for monitoring and predicting the dynamics of cyanobacteria and cyanotoxins in the reservoir. Microscopy confirmed the presence of potentially toxin-producing cyanobacteria at all sampling times, and qPCR detection showed the occurrence of both *mcyE* and *sxtA* in most samples. Concentrations of MC and STX were generally low (MC 0–1.54 µg L$^{-1}$; STX 0.03–0.21 µg L$^{-1}$). The highest MC level exceeded the recommended limit for human intake of 1 µg L$^{-1}$. The abundance of the *mcyE* and *sxtA* genes, as well as the toxin concentrations, were positively correlated with the biomass of *Phormidium* and *Raphidiopsis*. Among environmental variables, the abundance of potential toxic cyanobacteria was mainly affected by P limitation (high TN:TP ratios). Our data show that detection of the *mcyE* and *sxtA* genotypes serves as a useful and reliable predictor of toxin episodes but might be combined with chemical toxin detection to form an environmental toolbox for cyanotoxin monitoring.

**Keywords:** Lobo reservoir; cyanotoxins; *sxtA* gene; *mcyE* gene; *Phormidium*; *Raphidiopsis*

## 1. Introduction

Globally, eutrophication and climate change have caused an increased occurrence of cyanobacteria in many freshwaters [1]. Several species of cyanobacteria produce toxins that cause threats to water quality and pose risks to human and animal health [1,2]. To minimize these negative effects, monitoring and adequate management of cyanobacterial abundance are essential measures, particularly in freshwater systems used as sources of drinking water and for recreation. In natural environments, the conditions that promote the development of cyanobacterial blooms and induce toxin production are usually an interaction of a variety of factors [3], such as nutrient inputs [4,5], temperature [6,7], light intensities [8,9], pH [10,11], and hydrodynamics (i.e., water level, wind-driven currents, and stratification) [12,13]. For example, higher nutrient levels and higher temperatures may interact simultaneously with cyanobacterial growth and cyanotoxin production [14]. In this context, some studies suggest that climate change can intensify the effects of eutrophication in aquatic environments [15,16]. Therefore, the identification of major environmental

drivers and their consequences on the dominance of cyanobacteria and cyanotoxin production is important when deciding which management strategies should be applied, aiming to reduce the impacts of eutrophication and higher temperatures [1].

Among the toxins produced by cyanobacteria, microcystin (hepatotoxin) and saxitoxin (neurotoxin) are the most commonly reported worldwide [17]. Microcystin (MC) and saxitoxin (STX) are synthesized by gene clusters known as *mcy* and *sxt*, respectively [18–21]. Molecular characterization of the two gene clusters has enabled the application of selected *mcy* and *sxt* genes as targets for polymerase chain reaction (PCR) and later also for quantitative PCR (qPCR). Various *mcy* and *sxt* genes have been targeted by qPCR to quantify the abundance of specific MC and STX producers in natural environments [22]. For example, Al-Tebrineh et al. [23,24] developed a specific quantitative PCR method based on the *sxtA* gene to quantify saxitoxin-producing *Anabaena circinalis* among cyanobacteria in several Australian water bodies. In another study, the *mcyE* gene was targeted to quantify potential MC producers in Missisquoi Bay (QC, Canada), which is an area that supports important economic and recreational activities and serves as water intake for the drinking water treatment plant of surrounding towns [25]. Thus, the abundance of potentially toxic cyanobacteria in natural environments can be quantified by the amplification of genetic markers by qPCR, allowing differentiation between toxic and non-toxic but morphologically similar strains, e.g., within a species [22].

Lobo is a subtropical reservoir located in São Paulo State, Brazil. The main uses of water in the reservoir are the generation of hydroelectric energy, the supply of drinking water, the irrigation of plantations, and recreation, which contribute to the social and economic development of the surrounding area [26]. One episode of a major cyanobacterial bloom and an accompanying production of cyanotoxins occurred in July 2014 in the reservoir [27] and was caused by *Raphidiopsis raciborskii* (basionym *Cylindrospermopsis raciborskii*) [28]. During the bloom, the toxins saxitoxin and microcystin were detected in the reservoir. Reoccurring episodes of undesirable cyanotoxin concentrations may be expected since the reservoir is experiencing water quality deterioration due to the release of high loads of nutrients into the system [27,29].

Cyanobacterial genera with potential for the production of cyanotoxins have previously been identified in the phytoplankton community in Lobo reservoir [27,29,30], but molecular techniques for detection and quantification of microcystin- and saxitoxin-producing cyanobacteria in the reservoir have not been applied. Therefore, we examined the cyanobacterial community in the Lobo reservoir between May 2017 and January 2018 for the abundance of the *mcyE* and *sxtA* genes and quantified the MC and STX concentrations in the water. Aiming to assess the variables that might be applicable for monitoring the dynamics of potentially toxin-producing cyanobacteria in the reservoir, we examined correlations between environmental variables and cyanobacterial biomass, *mcyE* and *sxtA* genotypes, and concentrations of MC and STX.

## 2. Material and Methods

### 2.1. Study Sites and Sampling

Water samples were collected in the Lobo reservoir, located in São Paulo State, Brazil (Figure 1). The main characteristics of the reservoir and sampling sites are shown in Table S1. Samplings were carried out at riverine and dam zones in May, August, October 2017, and January 2018 at two depths (100% = surface; 1% = lower limit of the euphotic zone) determined by the photosynthetically active radiation (PAR, μmol photons m$^{-2}$ s$^{-1}$, Table S2) using a light sensor (LI-1400 DataLogger, sensitivity of 400–700 nm, LI-COR Biosciences, Lincoln, NE, USA) (*n* = 16).

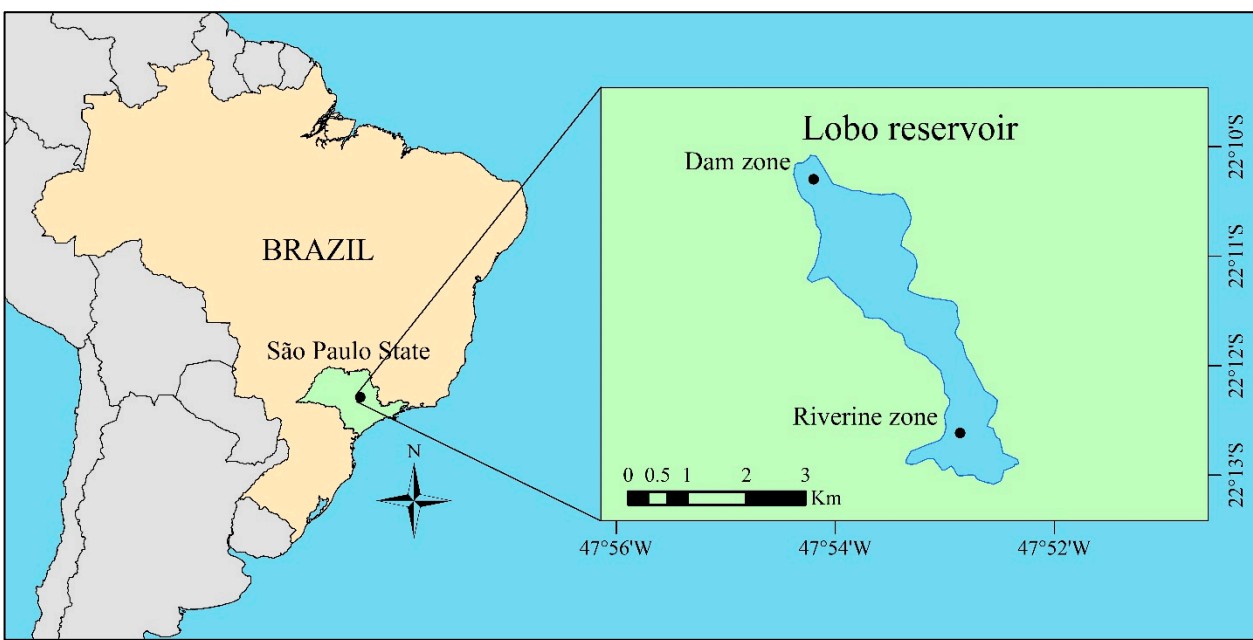

**Figure 1.** Sampling sites in Lobo reservoir, São Paulo State, Brazil: riverine zone (22°12′36.9″ S, 47°52′52.8″ W) and dam zone (22°10′18.5″ S, 47°54′11.2″ W).

## 2.2. Physicochemical Variables

The profiles of pH, dissolved oxygen (DO, mg L$^{-1}$), electrical conductivity (EC, μS cm$^{-1}$), turbidity (NTU), and water temperature (°C) were determined on-site using a HANNA probe (HANNA HI9829, Limena, Italy). Water transparency (m) was determined using a Secchi disk, and the lower limit of the euphotic zone ($Z_{eu}$, m) was determined as 1% of PAR. Water samples were filtered through 0.7 μm-pore-size glass fiber filters (GE Healthcare Life Sciences, Whatman, MA, USA). Chlorophyll *a* (chl *a*) in the material on the filters was extracted with ethanol 80% (*v/v*) and the extracts were analyzed as described by Nusch [31]. The filtrates were kept frozen at −20 °C until analysis of dissolved nutrients (soluble reactive phosphorus—SRP, nitrate—NO$_3^-$-N, nitrite—NO$_2$-N, and ammonium—NH$_4^+$-N). For unfiltered water, we quantified total Kjeldahl nitrogen (TKN) and total phosphorus (TP). Total nitrogen (TN) concentrations were estimated as the sum of TKN, NO$_2^-$-N, and NO$_3^-$-N. TN:TP molar ratios were also calculated. The nutrient analyses were carried out using spectrophotometric methods in triplicate (APHA) [32]. The trophic state index (TSI) of the sampling sites (riverine and dam zones) was calculated considering the annual geometric means of chl *a* and TP [33].

## 2.3. Microscopic Identification and Enumeration

For phytoplankton analysis, untreated water samples were Lugol-fixed for subsequent identification and quantification in sedimentation chambers using an inverted microscope (Olympus CK2) at 400× magnification following the Utermöhl (1958) [34] method. Phytoplankton taxa were identified according to specialized literature [35–40]. A counting limit was established through the species–rarefying curve until reaching 100 individuals (cell, filament, colony, and coenobium) of the most common species. Twenty organisms of each taxon with a relative abundance > 10% were measured to estimate the mean cell volume based on the geometric models [41,42]. For taxa with a relative abundance < 10%, mean cell volumes were obtained from specialized literature. Cell densities were converted to biovolume (mm$^3$ L$^{-1}$) by multiplying cell densities by the average taxa-specific cell volume. Phytoplankton biomass was estimated according to Wetzel and Likens [43], where 1 mm$^3$ L$^{-1}$ = 1 mg L$^{-1}$.

### 2.4. DNA Extraction and Quantitative Real-Time PCR

For the DNA extraction, water samples (volume of 500 mL) were immediately filtered through 0.22 µm-pore-size mixed cellulose ester membranes (GE Healthcare Life Sciences, Whatman, MA, USA), and the filters were kept frozen at −20 °C until processing. Total genomic DNA was extracted from the filters using the DNeasy PowerWater Kit (QIAGEN, Hilden, Germany) according to the manufacturer's instructions. DNA concentrations were quantified using a NanoDrop 2000c spectrophotometer (Thermo Scientific, Wilmington, NC, USA) [44]. The DNA purity was determined by the ratio of absorbance at 260 nm to absorbance at 280 nm. Further, the DNA quality was visualized on a 1% agarose gel stained with GelRed$^{TM}$ Nucleic Acid Gel Stain (Biotium Inc., Fremont, CA, USA) using a Bio-Rad Gel Doc™ 2000 gel documentation system (Bio-Rad Laboratories, Hercules, CA, USA).

Quantification of the *mcyE* gene was carried out using the CyanoDTec Toxin Gene Kit (Phytoxigene Inc., Akron, OH, USA). This kit is based upon molecular (DNA) technology (quantitative real-time PCR), which includes standards for the toxin gene target and validated primers and probes for the quantification of the toxin-producing genes in aquatic environments. A standard of known toxin gene copy was assayed in serial dilutions spanning four orders of magnitude (from $1 \times 10^5$ to $1 \times 10^2$ copies per reaction volume) to generate a standard curve for the MC target gene *mcyE* with an amplification efficiency of 96% ($R^2 = 0.99$; slope = −3.43). The analysis was performed according to the manufacturer's instructions using an AriaMx Real-Time PCR System (Agilent Technologies, Santa Clara, CA, USA) in a total volume of 20 µL per reaction. Each sample was run in duplicate. PCR was initiated with preheating at 95 °C for 2 min, followed by 40 PCR cycles, each consisting of 15 s at 95 °C and 45 s at 60 °C. Fluorescence measurement of the generated products was obtained at the end of each cycle at 60 °C. Gene copies in each reaction were calculated using the AriaMx Software version (1.3) (Agilent Technologies, Santa Clara, CA, USA) and back-calculated to copies mL$^{-1}$.

A qPCR assay for the *sxtA* gene using SYBR$^®$ Green was adopted for use in this study [45]. The qPCR assay was performed using the primer set *sxtA*-cyano-F (5′-TTATGAA-GCGTGCTGTCTGG-3′) and *sxtA*-cyano-R (5′-TCTGCCGACATGGAATACAC-3′). The qPCR product was a 153 bp fragment. The qPCR reactions were run in triplicate in an AriaMx Real-Time PCR System (Agilent Technologies, Santa Clara, CA, USA). The final volume for each qPCR reaction was 20 µL, containing 2 µL of template DNA from a standard or environmental sample, 0.4 µL of the *sxtA*-cyano-F and *sxtA*-cyano-R primers (10 pmol µL$^{-1}$), 10 µL of Brilliant III Ultra-Fast SYBR$^®$ Green qPCR Master Mix (Agilent Technologies, Santa Clara, CA, USA), and 7.2 µL of nuclease-free water (Sigma-Aldrich, San Luis, MO, USA). The qPCR assays were performed under the following cycling conditions: preheating at 95 °C for 3 min, followed by 40 cycles, each consisting of 30 s at 95 °C, 30 s at 65 °C, 30 s at 72 °C and 10 s at 77.5 °C, and a melt cycle of 30 s at 95 °C, 30 s at 65 °C, and 30 s at 95 °C. Fluorescence measurement of the generated products was obtained at the end of each cycle at 77.5 °C. Data analysis was performed by the AriaMx Software version (1.3) (Agilent Technologies, Santa Clara, CA, USA).

### 2.5. Cyanotoxin Analyses

Cyanotoxins were extracted from 50 mL unfiltered water samples by triplicate freeze-thaw cycles, including freezing at −80 °C for a minimum of 1 h and sonication at 37 °C for 5–10 min to lyse the cyanobacteria cells. Total saxitoxin (STX) and total microcystin (MC) concentrations were measured by enzyme-linked immunosorbent assay (ELISA) using commercial kits (Beacon Analytical Systems Inc., Saco, ME, USA) using procedures provided by the manufacturer.

### 2.6. Statistical Analysis

The obtained data were tested for normality using the Shapiro–Wilk test. A Mann–Whitney test was carried out to evaluate the differences between sites (riverine and dam zones) and depths (100% and 1% of PAR) considering the environmental variables and the biomass

of cyanobacteria. To evaluate the differences between months, a Kruskal–Wallis test was conducted considering the environmental variables and the biomass of cyanobacteria, and a Dunn's post hoc multiple comparison test was performed if significant differences were observed in the Kruskal–Wallis test ($p < 0.05$). A principal components analysis (PCA) with a correlation matrix was performed to summarize the environmental variability of the reservoir according to abiotic conditions registered during the studied period, characterizing the sampling sites, months, and depths, and identifying the variables that best differentiate them (strong correlations were those with $r \geq 0.5$ with the ordination axes 1 or 2). The relationships between cyanotoxins genes, cyanotoxins production, cyanobacteria biomass, and environmental variables were assessed by Spearman's rank-order correlations ($p < 0.05$). To obtain the significant factors that explained the occurrence of microcystin and saxitoxin in the reservoir, a simple linear regression analysis was conducted. Environmental variables were log(x + 1) transformed before analysis to meet the conditions of normality and homogeneity of variance in the residuals. Data were analyzed using STATISTICA version (13.5) (TIBCO Software Inc., Palo Alto, CA, USA) and PAST version (4.06) [46].

## 3. Results

### 3.1. Environmental Variables

Among environmental variables, significant seasonal changes were found for pH, electrical conductivity (EC), dissolved oxygen (DO), and water temperature (months; $p < 0.05$). As for spatial changes (riverine vs. dam zones), EC, turbidity, total phosphorus (TP), total nitrogen (TN), nitrate, and nitrite showed significant variations ($p < 0.05$). None of the environmental variables differed between depths ($p > 0.05$). Mean values, ranges, and statistical results are presented in Table S3.

The difference in environmental variables between the riverine and dam zones in the reservoir suggests an effect of the discharge of untreated wastewater in the riverine zone. In the riverine zone, mean values of EC (20.38 $\mu$S cm$^{-1}$), water turbidity (6.06 NTU), TP (25.92 $\mu$g L$^{-1}$), and soluble reactive P (2.36 $\mu$g L$^{-1}$), as well as TN (0.80 mg L$^{-1}$), were higher than in the dam zone. The high TN:TP ratio in both zones (64 in the riverine zone and 87 in the dam zone) indicates a general limitation by P in the reservoir. Concentrations of chl *a* (range 3.95 to 27.3 $\mu$g L$^{-1}$ in the reservoir) were occasionally higher in the dam zone, but no differences between the two zones were found during the studied period ($p > 0.05$). The trophic state index indicates that both zones can be characterized as mesotrophic.

### 3.2. Phytoplankton Community

Cyanobacteria were commonly occurring in the Lobo reservoir, but they did not make up the dominant phytoplankton group (Figure 2a). Dinoflagellates and cryptophytes were the most abundant groups of phytoplankton, followed by diatoms and green algae (Figure 2b). The highest biomass of cyanobacteria was observed in May 2017 in the dam zone (0.24 mg L$^{-1}$), corresponding to 12% of the total phytoplankton biomass. At the other sampling times, cyanobacteria made up less than 7% of the biomass (Figure 2b). Phytoplankton taxa identified in the reservoir by microscopy are listed in Table S4.

Fourteen species of cyanobacteria belonging to 10 genera were identified. Although in low proportion, potential microcystin (MC) and saxitoxin (STX)-producing organisms were detected in all samples by microscopic analysis (Figure 3), including *Aphanocapsa* spp., *Chroococcus* spp., *Microcystis protocystis*, *Pseudanabaena* spp., and *Synechocystis aquatilis* (all known as MC producers), *Aphanizomenon gracile*, *Geitlerinema amphibium*, *Raphidiopsis raciborskii* (known as STX producers), and *Phormidium* sp. (known as MC and STX producers) [47–51].

The occurrence of the cyanobacteria showed seasonal changes in some genera, but none of the cyanobacteria had significant vertical (surface vs. depth) or horizontal (riverine vs. dam) variations of their biomasses ($p > 0.05$). Cyanobacteria with a permanent occurrence included *Aphanocapsa* spp. and *S. aquatilis*, as well as *Chroococcus* spp. and *R. raciborskii*, but *Chroococcus* spp. and *R. raciborskii* both had a rather variable occurrence.

The highest biomass of *R. raciborskii* was observed in May and October 2017 in the dam zone. Among cyanobacteria with a seasonal variation in biomass were *Anathece* sp. (maximum biomass in October 2017 and January 2018) and *Aphanocapsa* spp. and *Pseudanabaena* spp. (highest biomasses in May 2017; *p* < 0.05). *Microcystis protocystis* was observed only in May and August 2017 in the dam zone. *Phormidium* sp. occurred only in May 2017 in both zones, while *A. gracile* was detected in August and October 2017 in the riverine zone and in May 2017 in the dam zone. *G. amphibium* was observed in May 2017 in both zones and in January 2018 in the dam zone (Figure 3, Table S4).

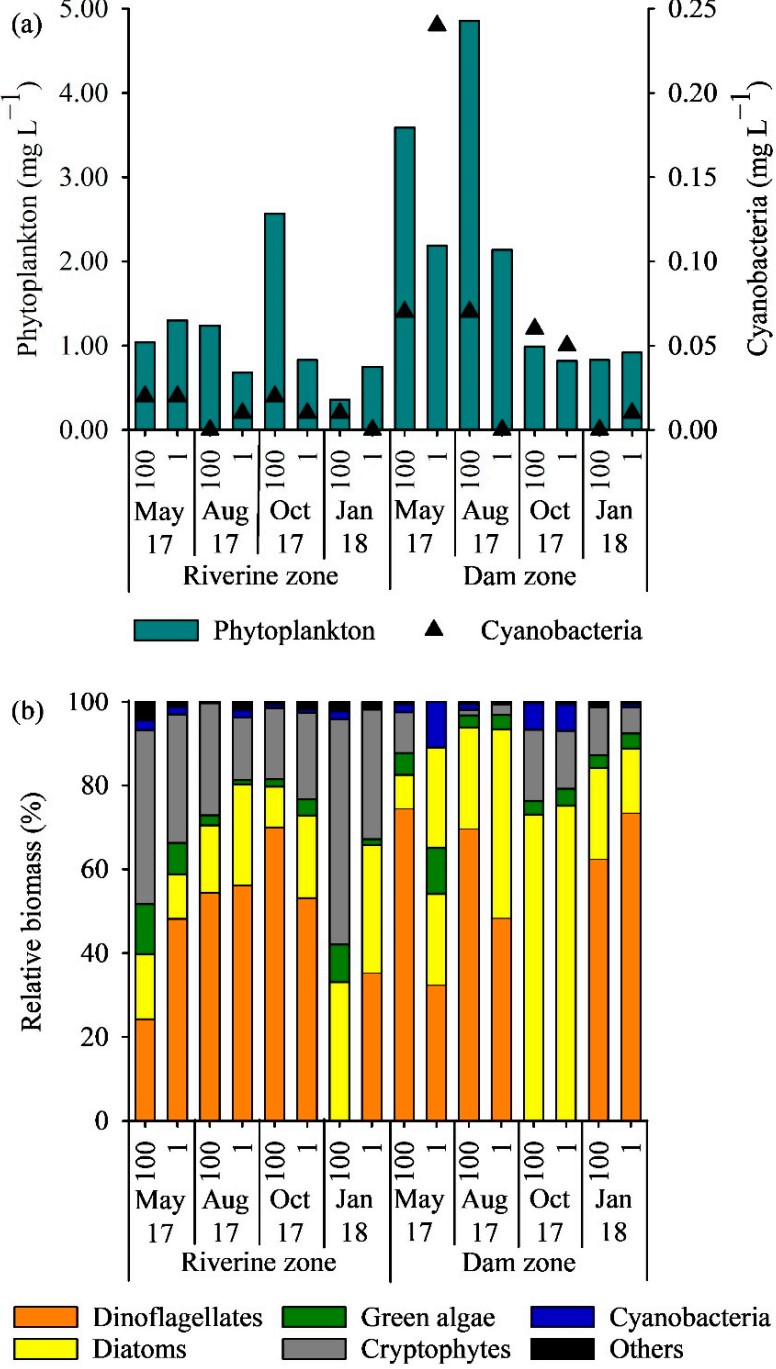

**Figure 2.** (**a**) Biomass (mg L$^{-1}$) of the total phytoplankton community and cyanobacteria and (**b**) relative biomass (%) of phytoplankton groups in the Lobo reservoir; 100 = surface; 1 = lower limit of euphotic zone.

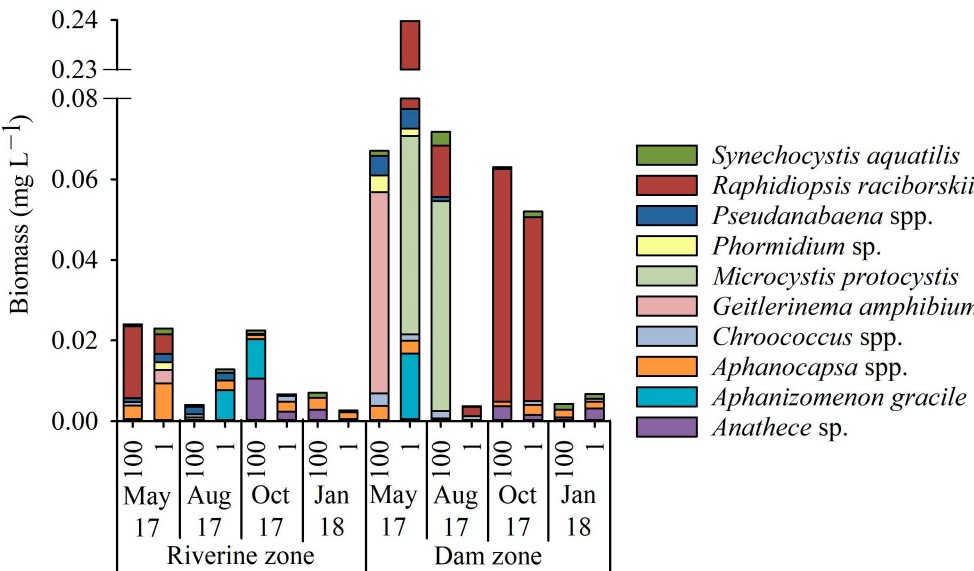

**Figure 3.** Biomass (mg L$^{-1}$) of cyanobacteria identified in the Lobo reservoir; 100 = surface; 1 = lower limit of euphotic zone.

### 3.3. Abundance of mcyE and sxtA Genes and Toxin Concentrations

The qPCR assay confirmed the presence of *mcyE* genotypes in the Lobo reservoir in May 2017 (riverine and dam zones) and in August 2017 (dam zone only). The number of the *mcyE* gene ranged from undetectable to $4.25 \times 10^3$ copies mL$^{-1}$ and total MC concentrations ranged from undetectable to 1.54 µg L$^{-1}$ (Figure 4) and correlated positively with the *mcyE* gene (*rho* = 0.75, *p* < 0.001) (Table 1). Microcystin concentrations in the Lobo reservoir were positively correlated with the biomass of *Phormidium* sp. (*rho* = 0.66, *p* = 0.005). Although no correlations between concentrations of MC and biomass of the potential MC producers *Aphanocapsa* spp., *Chroococcus* spp., *M. protocystis*, *Pseudanabaena* spp., and *S. aquatilis* were observed, the possibility that these cyanobacteria were also producing MC in the Lobo reservoir should not be ruled out since they were present in the reservoir when MC was detected.

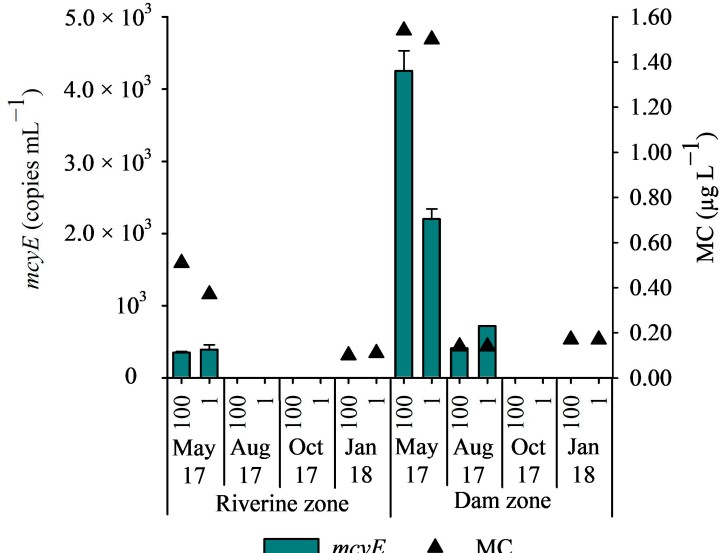

**Figure 4.** Abundance of *mcyE* gene (copies mL$^{-1}$) and microcystin concentration (MC, µg L$^{-1}$) in the Lobo reservoir. Error bars indicate standard deviations of triplicate samples; 100 = surface; 1 = lower limit of euphotic zone.

**Table 1.** Spearman rank correlation coefficients (*rho*) between environmental variables, cyanobacterial biomass, microcystin concentration, *mcyE* gene numbers, saxitoxin concentrations, and *sxtA* gene numbers in Lobo reservoir (*n* = 16). Significant correlations (*p* < 0.05) are highlighted in bold.

| | EC | NO$_2$$^-$-N | TP | SRP | Chl *a* | MC | *mcyE* | STX | *sxtA* |
|---|---|---|---|---|---|---|---|---|---|
| Cyanobacterial biomass | **−0.59** | **−0.53** | **−0.55** | −0.08 | 0.27 | 0.32 | 0.49 | **0.55** | **0.69** |
| MC | −0.11 | **−0.62** | −0.36 | 0.33 | **0.71** | - | **0.75** | - | - |
| *mcyE* | **−0.62** | **−0.51** | **−0.65** | 0.14 | **0.75** | **0.75** | - | - | - |
| STX | −0.13 | −0.37 | −0.27 | **0.51** | 0.20 | - | - | - | **0.78** |
| *sxtA* | −0.40 | **−0.52** | −0.36 | 0.28 | 0.21 | - | - | **0.78** | - |

EC = electrical conductivity; NO$_2$$^-$-N = nitrite; TP = total phosphorus; SRP = soluble reactive phosphorus; Chl *a* = chlorophyll *a*; MC = microcystin; STX = saxitoxin.

The *sxtA* gene was detected by qPCR in all samples but showed a temporal variation in abundance (*p* = 0.017). The number of copies of the *sxtA* gene in the reservoir during the studied period ranged from $8.79 \times 10^1$ to $2.34 \times 10^4$ copies mL$^{-1}$. The highest and lowest numbers were measured in May 2017 in the dam zone and in January 2018 in the riverine zone, respectively (Figure 5). Saxitoxin (STX) concentrations varied significantly between months (*p* = 0.022). When STX was detected in the water, the concentrations varied between 0.03 and 0.21 µg L$^{-1}$ (Figure 5) and correlated positively with the *sxtA* gene abundance (*rho* = 0.74, *p* < 0.001) (Table 1). The presence of STX correlated positively with the biomasses of *Phormidium* sp. (*rho* = 0.64, *p* = 0.008) and *R. raciborskii* (*rho* = 0.58, *p* = 0.02). No correlations were observed between concentrations of STX and biomasses of *A. gracile* and *G. amphibium*, but it cannot be excluded that these potential STX producers were also responsible for the saxitoxin production in the reservoir.

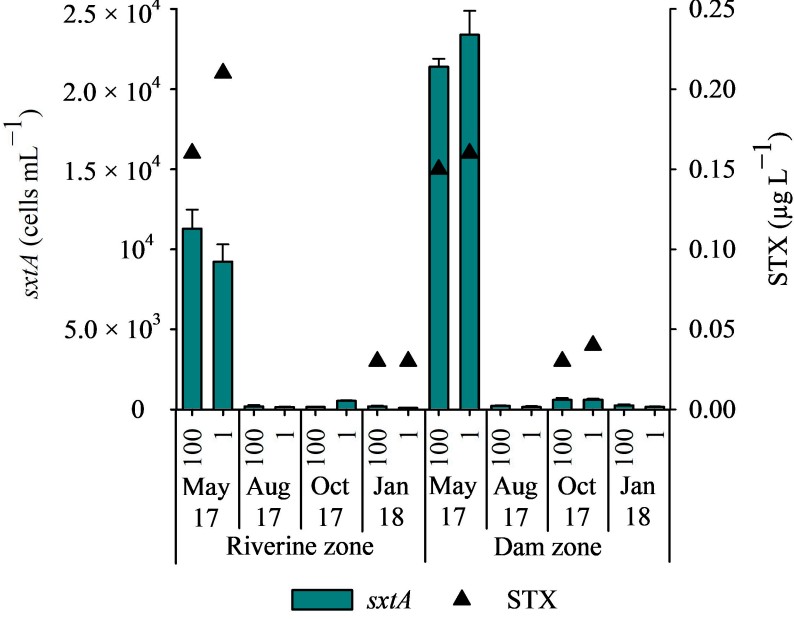

**Figure 5.** Abundance of the *sxtA* gene (cells mL$^{-1}$) and saxitoxin concentration (STX, µg L$^{-1}$) in the Lobo reservoir. Error bars show standard deviations of triplicate samples; 100 = surface; 1 = lower limit of euphotic zone.

### 3.4. Relationship between Environmental Variables, Cyanobacteria and Cyanotoxins

A principal component analysis (PCA) showed that the 19 initial environmental variables could be reduced to only 9 variables that accounted for 72.81% of the total variance of the two first components (PC 1 = 53.80% and PC 2 = 19.01%) (Figure 6). The main variables that positively correlated to the PC 1 axis were MC (*r* = 0.98), *mcyE* gene (*r* = 0.89), saxitoxin (*r* = 0.80), *sxtA* gene (*r* = 0.97), and chl *a* (*r* = 0.89). For the PC 2 axis, electrical conductivity (EC; *r* = 0.63), TP (*r* = 0.69), and soluble reactive P (*r* = 0.86) showed

positive correlations. Riverine samples from October and January appear to be similar for high nutrient concentrations (TP and nitrite) and EC. In May 2017, the riverine and dam zones were characterized by high levels of STX, *sxtA*, MC, *mcyE,* and chl *a*.

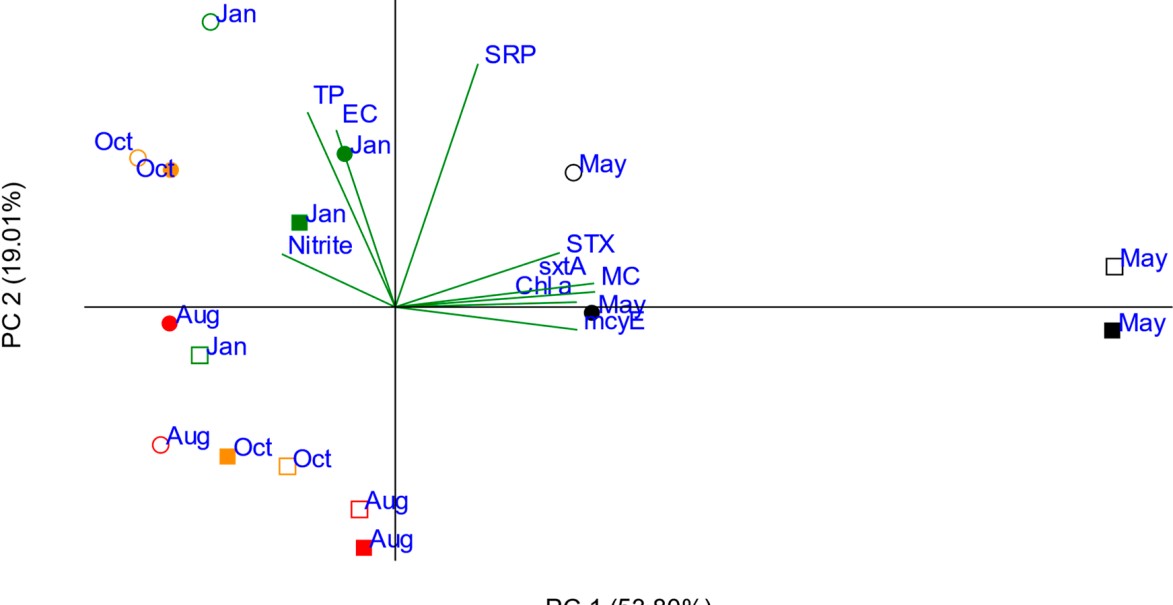

**Figure 6.** Ordination biplot by principal component analysis (PCA) of sample units generated from nine environmental variables in the Lobo reservoir. The sampling times were May 2017 (black), August 2017 (red), October 2017 (orange) and January 2018 (green). The riverine zone is represented by circles (●○), and dam zone is represented by squares (■□). Filled dots (●) or squares (■) represent the depth 100 (surface), while empty (○) dots and squares (□) represent the depth 1 (lower limit of euphotic zone). TP = total phosphorus; EC = electrical conductivity; SRP = soluble reactive phosphorus; STX = saxitoxin; MC = microcystin; Chl *a* = chlorophyll *a*.

The correlation analyses in Table 1 indicated that different variables influenced the abundance of the *mcyE* and *sxtA* genotypes as well as concentrations of MC and STX. MC concentrations and *mcyE* gene numbers were positively correlated with chl *a* but negatively correlated with nitrite. The *mcyE* gene number was negatively correlated with EC and TP. STX concentrations and *sxtA* gene numbers were positively correlated with cyanobacterial biomass. STX concentrations were also positively correlated with soluble reactive P, while the *sxtA* gene was negatively correlated with nitrite.

A linear regression analysis suggested an acceptable estimate of the MC concentration by both chl *a* ($R^2 = 0.80$, $p < 0.001$) and *mcyE* gene number ($R^2 = 0.86$, $p < 0.001$). Regarding STX, the best predictor was the *sxtA* gene number ($R^2 = 0.68$, $p < 0.001$). Despite the moderate correlation observed between cyanobacterial biomass and STX, regression analysis indicated that cyanobacterial biomass was not a good fit for the prediction of STX (Table 2).

**Table 2.** Linear regression of microcystin and saxitoxin concentrations to other measured variables ($n = 16$).

| Variables | Microcystin | | Saxitoxin | |
|---|---|---|---|---|
| | $R^2$ Adj | *p* Value | $R^2$ Adj | *p* Value |
| Chlorophyll *a* | 0.80 | < 0.001 | - | - |
| *mcyE* | 0.86 | < 0.001 | - | - |
| Cyanobacterial biomass | - | - | 0.23 | 0.04 |
| *sxtA* | - | - | 0.68 | < 0.001 |

## 4. Discussion

### 4.1. Cyanobacteria and Cyanotoxins in the Reservoir

The water quality in the Lobo reservoir has been characterized as "compromised" due to the discharge of nutrients from agriculture, livestock, residential areas, and various diffuse sources [29]. The effects of this eutrophication were studied in July 2014, when a cyanobacterial bloom dominated by *Raphidiopsis raciborskii* was observed in the reservoir [27]. During the bloom, MC and STX were detected in the water, but no concentrations were reported [27]. The authors emphasized the need for future monitoring of cyanobacteria and cyanotoxins due to the continuous discharge of phosphorus and nitrogen. In the present study, both the riverine and the dam sampling sites were classified as mesotrophic (TP < 37.85 µg L$^{-1}$ and chl *a* < 27.30 µg L$^{-1}$), and the average TN:TP ratios (64:1 in the riverine zone and 87:1 in the dam zone) indicate a limitation by phosphorus at both sites. During our study, the phytoplankton community was mainly characterized by the presence of dinoflagellates, diatoms, cryptophytes, and chlorophytes. Cyanobacteria occurred at all sampling times, but at a low biomass. However, among the ten cyanobacterial genera observed in the reservoir, nine are known as potentially cyanotoxin-producing organisms.

Variations in concentrations of MC and STX in the lower stratum of the euphotic zone and at the surface of the reservoir did not show clear trends. Possibly, wind action and the shallow depth at the sampling sites (max. 2.5 m and 11.5 m depths in the riverine and dam zones, respectively) (Table S1) resulted in the mixing of the water column. Statistical analyses indicated that MC production could be related to the genus *Phormidium* and that STX production could be associated with the genera *Phormidium* and *Raphidiopsis* in the reservoir. However, since other potential toxin-producing genera of cyanobacteria also occurred in the water (Figure 3, Table S4), it cannot be confirmed that these genera actually produced toxins in the reservoir.

Although the cyanobacterial biomass was low and no bloom episodes occurred in the reservoir during the studied period, this does not exclude the production and release of cyanotoxins in the water. In a study on microcystin in Lake Vancouver, where the abundance of *Microcystis* sp. rarely exceeded one percent of the cyanobacteria, Lee et al. [4] observed that the majority of the *Microcystis* population contained the *mcyE* gene, and concentrations of MC above the WHO guidelines for drinking water were repeatedly measured. In our study, concentrations of MC exceeded the allowed limit of 1.0 µg L$^{-1}$ for human consumption (Brazilian Ministry of Health; Ordinance 2914/2011) [52] in May 2017 in the dam zone. STX was detected in half of the Lobo water samples, but the concentrations were below the level indicated for drinking water (3.00 µg L$^{-1}$), as also reported for the reservoir by Tundisi et al. [27].

According to current Brazilian legislation [52], concentrations of MC and STX should be quantified weekly, but only if the cyanobacterial density exceeds 20,000 cells mL$^{-1}$. However, the density of cyanobacteria should not be the only parameter for verifying the potential for cyanotoxin production, since the abundance of cyanobacteria does not identify the toxic potential of a cyanobacterial community, as our results indicate. Further, microscopic analyses are time-consuming and therefore microscopic counts may not be immediately available during bloom episodes. According to our results, the abundance of the *mcyE* and *sxtA* genes correlated significantly with concentrations of MC and STX, respectively, and *mcyE* and MC also correlated with chl *a* concentrations. Therefore, to improve fast monitoring for toxin production and to better assess the actual situation of a water body with respect to the occurrence of cyanotoxins, we suggest the application of quantitative molecular methods targeting toxin-encoding genes, such as the qPCR procedure. By combining this approach with quantification of chl *a* concentration, this would allow faster estimates of potential toxin episodes than time-consuming microscopy. If possible, it is also important to determine the toxin concentration using analytical methods to verify the actual toxicity of the water.

### 4.2. Environmental Variables and Cyanotoxins

In the Lobo reservoir, our results showed that concentrations of MC and numbers of the *mcyE* and *sxtA* genes were negatively correlated with nitrite concentrations. Concentrations of STX correlated positively with soluble reactive P, while the *sxtA* gene abundance was negatively related to total P. Previous studies demonstrated that concentrations of N and P are significant parameters affecting microcystin and saxitoxin production, as well as the abundance of toxic genotypes [4,5,53]. However, the importance of each of these parameters appears to vary strongly between different environments. Nutrient control of MC production was demonstrated in laboratory experiments by Pimentel and Giani [48], who observed that deprivation of N and P promoted the *mcyD* gene transcription, possibly due to oxidative stress during the nutrient limitation. Supporting this, an increased transcription of the *mcy* gene was also measured during nitrate starvation by Ginn et al. [54]. For STX, Cirés et al. [7] measured an increased production by *Aphanizomenon gracile* when dissolved inorganic N was depleted. Similar results on the influence of N on the production of STX have been reported in other studies [55,56].

In the Lobo reservoir, the low availability of soluble reactive P ($< 3.70\ \mu g\ L^{-1}$) may have affected the positive correlation between this source of P and concentrations of STX. Vargas et al. [5] observed that low levels of P ($\leq 3.30\ \mu g\ L^{-1}$) stimulated the production of STX by *R. raciborskii*, and this might indicate a survival strategy and adaptation to P-limited environments. Considering that the soluble reactive P level in our study was similar to that used by Vargas et al. [5], we tend to conclude that the low availability of P in the Lobo reservoir may have affected STX production by cyanobacteria. Furthermore, although *R. raciborskii* can dominate at both low and high TN:TP ratios, the production of saxitoxin is enhanced at high TN:TP ratios (> 40:1) [57]. In the Lobo reservoir, the TN:TP ratios varied between 30:1 (riverine zone) and 124:1 (dam zone), which may explain the occurrence of STX despite the low cyanobacterial biomass observed in the reservoir.

The abundance of the *mcyE* gene was significantly and negatively related to electrical conductivity (EC) in the reservoir. Tao et al. [58] examined key factors affecting the cell-specific content of MC in *Microcystis* sp. in Lake Taihu (China), and they concluded that EC explained 21% of the variation in MC content and was the highest weighted parameter in the applied model. Since the EC parameter reflects the ability of water to conduct electricity, including the concentration of inorganic ions, e.g., nutrients, the authors explained that low values of EC can be an indication of suboptimal conditions for growth (due to fewer nutrients), which may promote MC production as a response to the nutrient limitation.

### 4.3. Monitoring of Toxic Cyanobacteria

Although concentrations of MC and STX covaried with the abundance of the respective functional genes, *mcyE* and *sxtA*, an unexpected observation was found in January 2018 when MC was measured in the water, while no copies of *mcyE* were detected. Considering that microcystins can remain in the water for several weeks [59,60], this discrepancy was probably caused by the persistence of MC in the reservoir, even after the disappearance of the MC-producing cyanobacteria, as also reported by Nimptsch et al. [61]. Regarding STX, the detection of the *sxtA* gene in all samples indicated the presence of potentially saxitoxin-producing cyanobacteria during the entire studied period, yet no STX was measured in half of the water samples. This might be due to either down-regulation of the *sxt* gene expression [62] or that the *sxtA*-carrying cells detected by qPCR did not produce sufficient saxitoxin to be measured by the ELISA method (detection limit of $0.02\ \mu g\ L^{-1}$).

The direct relationship between cyanotoxin concentrations and gene copy numbers found in most cases in the reservoir agrees with reports from other studies [4,23–25]. However, in other cases, no correlations were found in natural waters [63–65]. This inconsistency may be attributed to several methodological variations, e.g., DNA extraction methods, target genes and primers for qPCR, and the method applied for cyanotoxin quantification, e.g., ELISA, HPLC, and LC-MS/MS, but also whether intra-or extracellular fractions are measured, as reported by Guedes et al. [64] and Pacheco et al. [22]. These authors

raised concern regarding the estimation of cyanotoxin concentration in environmental samples from the quantification of toxic positive genotypes since the factors stated above can contribute to discrepancies.

The present finding of a positive correlation between concentrations of chl *a* and MC concentrations and *mcyE* gene copy numbers agrees with observations from other studies, suggesting that monitoring of chl *a* could serve as a first warning sign for the presence of microcystin in water bodies [3,15,66]. Cunha et al. [67] used a 5-year time series of MC concentrations in six subtropical reservoirs to investigate the main factors related to MC concentrations and estimate thresholds for predicting toxin production. The authors found that for MC $\geq$ 0.10 µg L$^{-1}$, the chl *a* threshold was 13.00 µg L$^{-1}$, while the threshold for MC $\geq$ 1.00 µg L$^{-1}$ was 39.80 µg chl *a* L$^{-1}$. In the Lobo reservoir, the maximum MC concentration (1.54 µg L$^{-1}$) occurred when chl *a* was below 28.00 µg L$^{-1}$. Since the study of Cunha et al. [67] included a larger set of samples, and considering the limitations of our study (a relatively small number of samples, one reservoir, and low cyanobacterial biomass), the difference between our result and these authors' chl *a* thresholds for prediction of toxin production may be expected. Thus, long-term monitoring appears needed to establish an MC vs. chl *a* relationship. Although chl *a* might be used as an indirect indicator of potentially harmful blooms, correlations between toxic cyanobacteria and chl *a* may vary since cyanobacteria have photosynthetic active pigments other than chl *a*, and other phytoplankton groups might also contribute chl *a* [67]. Therefore, although chl *a* was strongly associated with MC production in the Lobo reservoir ($R^2 = 0.80$, $p < 0.001$), caution must be taken when using this variable alone for the prediction of the toxic MC genotypes.

Indicators for the prediction of MC production may vary between field sites with changeable conditions and controllable laboratory conditions. This was examined by Ngwa et al. [25], who observed that for laboratory cultures of *Microcystis* and *Planktothrix*, the chl *a* concentration was the best predictor of MC concentrations, followed closely by *mcyE* gene copy numbers and microscopic cell counts, whereas in environmental samples, *mcyE* gene copy numbers were the best indicators of MC concentrations, especially in water bodies with mixed populations of toxic and non-toxic cyanobacteria. Similarly, Davis et al. [6] reported that the abundance of the *mcyD* gene in *Microcystis* was significantly correlated with microcystin concentrations in three lakes and one pond in the USA, and this gene was a better predictor of MC concentrations than total cell counts or chl *a* concentrations. These findings reinforce the results found in our study since the *mcyE* abundance appeared to be the best predictor of potentially microcystin-producing cyanobacteria in the Lobo reservoir ($R^2 = 0.86$, $p < 0.001$). Thus, our results support the observations by Ngwa et al. [25] and Davis et al. [6] that the application of qPCR may serve as a valuable early monitoring tool that can reliably detect the presence of potentially toxic strains in natural waters. As stated above, chl *a* can be useful as a complementary tool to the qPCR approach since it is a fast estimate, but caution must be taken when using only this variable for the prediction of toxin production.

As for STX, a moderate but significant correlation was observed between STX concentrations and cyanobacterial biomass ($rho = 0.55$, $p < 0.05$). However, a linear regression model calculation indicated that cyanobacterial biomass was not a good fit for the estimation of total STX in the reservoir ($R^2 = 0.23$, $p = 0.04$). The best predictor of potentially saxitoxin-producing cyanobacteria was the abundance of the *sxtA* gene ($R^2 = 0.68$, $p < 0.001$). Therefore, we argue that an estimate of the *sxtA* gene abundance does not only detect the occurrence of potentially saxitoxin-producing cyanobacteria; it can also serve as an early warning tool for STX concentrations in the water, as stated in other studies [23,24]. However, although the qPCR data from the Lobo reservoir showed that the *mcyE* and *sxtA* genes can be used as indicators of microcystin and saxitoxin concentrations, this relationship may not be universal since local environmental factors may affect the toxin's production. Hence, to confirm the presence of cyanotoxins, chemical analytical methods, such as ELISA and LC-MS, are still needed.

## 5. Conclusions

Potentially toxin-producing cyanobacteria were permanently present in the Lobo reservoir, but their biomasses varied due to changes in environmental conditions. The abundance of these cyanobacterial strains implies the risk of a persistent occurrence of cyanotoxins in the water. Concentrations of MC and STX varied temporally, but the highest concentrations occurred during the dry winter season. The toxin concentrations were typically low, but in May 2017 in the dam zone, the MC concentrations exceeded the limit for human health of 1.00 µg $L^{-1}$, despite the occurrence of low cyanobacterial biomass. These results stress the importance of establishing improved monitoring of the quality of water used for human consumption since even low concentrations of MC may threaten human health [68].

In the Lobo reservoir, the presence of MC, STX, and toxic genotypes was influenced by several environmental variables, but their occurrence appeared to be mainly affected by the phosphorus limitation (high TN:TP ratios). Although nutrient limitation may restrict cyanobacterial growth, our results show that nutrient limitation may also cause an increase in toxin production. To better understand mechanisms between nutrient availability, cyanobacterial growth, and toxin production, we recommend that more studies with regular sampling over a longer period and in several water bodies (including this one) be conducted to determine relations between the growth of toxic cyanobacteria and their toxin production under different environmental conditions.

The current Brazilian legislation requires that concentrations of cyanotoxins be quantified weekly, but only if the cyanobacterial density exceeds 20,000 cells $mL^{-1}$, e.g., during cyanobacterial blooms. However, cyanobacterial density should not be the only criterion for verifying the potential for cyanotoxins production in freshwater systems, since cell counts cannot identify toxic vs. non-toxic species. Therefore, and considering the results from our study, we recommend using the qPCR approach to estimate the presence of potentially toxin-producing cyanobacteria in freshwaters. This will contribute to a reliable and fast risk evaluation to protect water users from hazards associated with toxic cyanobacteria. However, the actual presence of specific cyanotoxins and their concentrations still depend on analytical methods. Therefore, an environmental toolbox for monitoring cyanotoxins, such as MC or STX, should include qPCR detection of the functional genes in conjunction with ELISA or LC-MS analysis to provide better early warning as well as actual toxicity risk.

**Supplementary Materials:** The following supporting information can be downloaded at: https://www.mdpi.com/article/10.3390/environments10080143/s1, Table S1: Main characteristics of the Lobo reservoir; Table S2: Photosynthetically active radiation (PAR, µmol photons $m^{-2}$ $s^{-1}$) at surface (100% PAR) and at lower limit of euphotic zone (1% PAR) in the Lobo reservoir; Table S3: Environmental variables summarized as the mean values and ranges, and statistical results (*p* values) of non-parametric Mann-Whitney test for spatial differences (riverine and dam zone) and non-parametric Kruskal-Wallis for temporal differences (months) of the environmental variables in Lobo reservoir; Table S4: List of the phytoplankton taxa identified in Lobo reservoir.

**Author Contributions:** M.d.A.B.M. conceptualization, methodology, formal analysis, investigation, writing—original draft, writing—review and editing, visualization; R.d.A.M.R. investigation, writing—review and editing; R.P. methodology, investigation, writing—review and editing; N.O.G.J. writing—review and editing; M.d.C.C. conceptualization, methodology, writing—review and editing, supervision, project administration, funding acquisition. All authors have read and agreed to the published version of the manuscript.

**Funding:** This research was funded by Fundação de Amparo à Pesquisa do Estado de São Paulo (FAPESP)—Brazil, grant number 2016/09405-1. M.d.A.B.M. was funded by FAPESP, grant numbers 2015/21191-4 and 2018/00394-2. R.d.A.M.R. was funded by Coordenação de Aperfeiçoamento de Pessoal de Nível Superior (CAPES)—Brazil, grant number (finance number) 001, and Conselho Nacional de Desenvolvimento Científico e Tecnológico (CNPq)—Brazil, grant number 142176/2016-8.

**Institutional Review Board Statement:** Not applicable.

**Informed Consent Statement:** Not applicable.

**Data Availability Statement:** The data presented in this study will be made available upon request to the corresponding author.

**Conflicts of Interest:** The authors declare no conflict of interest.

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
