# Peer review of "Prediction of Cyanotoxin Episodes in Freshwater: A Case Study on Microcystin and Saxitoxin in the Lobo Reservoir, São Paulo State, Brazil"

_environments, doi:10.3390/environments10080143_

Round 1

Reviewer 1 Report

The manuscript is well written compiling various environmental and laboratory obtained data, and the final conclusion is also well mentioned. However, a few minor corrections are required .

Review comments on “Prediction of cyanotoxin episodes in freshwater: A case study on microcystin and saxitoxin in the Lobo reservoir, São Paulo State, Brazil”

Line89: “photosynthetically active radiation (PAR, μE m−2 s−1) using” …provide the light intensity range information and change the unit to “μmol photons m−2 s−1” as standard microbiological practice.

Line 116: “Phytoplankton taxa were identified according to specialized literature” Provide all references used for identification of various taxa in this study.

Line172: “enzyme-linked immune-assay (ELISA),” check for correctness

Lines400, 404, 536; Some random errors like “Clique ou toque aqui 400 para inserir o texto.”

its ok

Author Response

Replies to reviewer 1.

Reviewer: Line 89: “photosynthetically active radiation (PAR, μE m−2 s−1) using” …provide the light intensity range information and change the unit to “μmol photons m−2 s−1” as standard microbiological practice.

Authors: We have changed the unit from μE m−2 s−1 to μmol photons m−2 s−1 as requested by the reviewer. Besides that, we have added the light intensity data in the Table S2 in the Supplementary data.

Reviewer: Line 116: “Phytoplankton taxa were identified according to specialized literature” Provide all references used for identification of various taxa in this study.

Authors: We have provided all the specialized literature used for the taxonomic classification of phytoplankton organisms (references 35 to 40).

Reviewer: Line 172: “enzyme-linked immune-assay (ELISA),” check for correctness.

Authors: We have changed “enzyme-linked immune-assay (ELISA)” by “enzyme-linked immunosorbent assay (ELISA)”.

Reviewer: Lines 400, 404, 536: Some random errors like “Clique ou toque aqui para inserir o texto.”

Authors: We have excluded these typos from the text.

Reviewer 2 Report

Overall, the MS is well written with a significant focus. However, the topic and methods are not new here, as there are several articles that are quite similar, but these publications are still important for local monitoring and mitigation, so while it is not crucially new and innovative, it is still imperative.

Some minor changes are pointed out in the ms pdf.

Author Response

Replies to reviewer 2.

Reviewer: Some minor changes are pointed out in the ms pdf.

Authors: We have done the changes suggested by the reviewer in the manuscript. In addition, minor corrections of the text has been performed.

Reviewer 3 Report

The subject of the study corresponds to the scope of the journal "Environments".

The aim of this study was  to assess the different parameters that might be suitable for monitoring dynamics of potentially toxin-producing cyanobacteria in the Lobo reservoir, São Paulo State, Brazil.

The Authors have studied correlations between several environmental water parameters and cyanobacterial biomass, mcyE and sxtA genotypes and concentrations of cyanotoxines MC and STX.

The Materials and Methods section is written in sufficient detail and clearly.

The results are presented neatly and provided with informative figures and tables.

In the Discussion section the Authors have considered the main aspects of the results obtained in comparison with the results of other authors.

The conducted research was performed soundly. The Authors used well-known methods. The Authors managed to find correlations between the formation of cyanotoxins and various environmental parameters. The results obtained can be useful for analyzing situations with the proliferation of cyanobacteria and the presence of cyanotoxins in other reservoirs.

A few minor comments are below.

1) line 396 "rarely" has to be written with lowercase letter  "r".

2) Lines 400-401 and Line 536.  Please, remove the phrase "Clique ou toque aqui para inserir o texto.."

3) No deviations are indicated in Figure 2a. 

Author Response

Replies to reviewer 3.

Reviewer: 1) line 396 "rarely" has to be written with lowercase letter “r”.

Authors: We have corrected this writing in the text.

Reviewer: 2) Lines 400-401 and Line 536.  Please, remove the phrase "Clique ou toque aqui para inserir o texto."

Authors: We have excluded these typos from the text.

Reviewer: 3) No deviations are indicated in Figure 2a.

Authors: Figure 2a presents data from microscopic analysis of the phytoplankton community. This analysis was done without replicates. It is our own experience that variation between replicates of phytoplankton counts are 5-10%. Replicate analyses are typically not performed in phytoplankton counting, since the microscopy counting is very time-consuming (95 different species or genera were identified in our study), and since the variation among multiple counts usually is low. In our study, an expected variation of 5-10% between replicates would not affect correlations or conclusions. Details of the microscopy and conversions used are shown in M&M, section 2.3.

Reviewer 4 Report

Determining the relationship between the growth of toxic cyanobacteria and their production of toxins under various environmental conditions is an important area of research in freshwater ecosystems. The authors propose to expand the range of environmental tools for monitoring cyanotoxins to provide better early warning as well as the actual risk of toxicity.

A small note:

Line 536 is probably a typo: "Clique ou toque aqui para inserir o texto"

Author Response

Replies to reviewer 4.

Reviewer: Line 536 is probably a typo: "Clique ou toque aqui para inserir o texto".

Authors: We have excluded these typos from the text.